# Multi-carrier information hiding based on projection-driven vertex embedding in 3D models

**Yuefa Ou[1,2], Jie Ke[3], Mingkun Yang[3], Haobo Chen[4], Zhuoyi Dan[4], Chenglong Zhao[1,2]***

**1** Guangxi Key Laboratory of Ocean Engineering Equipment and Technology, Beibu Gulf University, Qinzhou, China, **2** School of Mechanical and Marine Engineering, Beibu Gulf University, Qinzhou, China, **3** School of Computer Science and Engineering, Guilin University of Aerospace Technology, Guilin, Guangxi, China, **4** School of Information Engineering, Chang'an University, Xian, China

* zhaochenglong@bbgu.edu.cn

## Abstract

To enhance the robustness of single 3D model carriers in information hiding, this paper proposes a multi-carrier steganography algorithm based on vertex projection of 3D models. The algorithm improves the embedding capacity and attack resistance by fusing multiple 3D models into a geometrically invariant space using centroid coincidence and tangent plane projection. Secret information is embedded by adjusting the position of central vertices in the projection plane of their 1-ring neighborhoods. Experimental results demonstrate that the proposed method achieves strong robustness against translation, rotation, simplification, random noise, and shear attacks. Specifically, the proposed algorithm achieves a peak SNR of 47.87 dB on the xyz-rgb_dragon model, significantly outperforming other algorithms, and maintains a lower BER under various attack intensities—for instance, less than 0.1 under 30% simplification and 0.001 random noise. These results confirm the superior invisibility and robustness of the proposed multi-carrier information hiding scheme.

## 1. Introduction

With the rapid development of network communication and the widespread use of smart devices in daily life, the transmission of information over the internet has become faster and more efficient, making communication between individuals more convenient. However, at the same time, due to the ease with which data can be copied, edited, and spread during transmission, multimedia communication faces significant security risks. Therefore, the security of secret information transmission urgently needs to be addressed. Information hiding technology-based secret information transmission can effectively solve this problem. As an important method of information hiding, 3D model information hiding technology holds great potential in addressing the security risks of multimedia communication. By embedding secret information into 3D models, it is possible to achieve covert transmission of secret information while maintaining data integrity and confidentiality during the transmission process.

**Data availability statement:** All relevant data are within the paper and its Supporting Information files.

**Funding:** This study was supported by: the Guangxi Natural Science Foundation Joint Special Project ("Beibu Gulf University Special Project"), Grant No. 2025GXNSFHA069099, received by Dr. Yuefa Ou; and the 2024 Guangxi Universities Enhancement of Research Capability for Young and Middle-aged Teachers Project, Grant No. 2024KY0447, received by Dr. Chenglong Zhao.

**Competing interests:** The authors have declared that no competing interests exist.

Zein et al. [1] proposed a non-blind robust hiding algorithm for 3D mesh models using K-means clustering technology. The insertion of secret information is achieved by modifying the vertex positions without causing noticeable distortion to the 3D mesh model. Wang et al. [2] used ordered statistics extraction and tracking algorithms to identify feature vertices of 3D models. By combining the distribution characteristics of feature vertex numbers, they performed information hiding, which is suitable for 3D models with multiple vertices. Tsai et al. [3] employed a fixed threshold to control the maximum embedding capacity of each vertex while reducing model distortion. They adaptively embedded different amounts of secret information based on the surface characteristics of each vertex, resulting in lower model distortion. Ren Shuai et al. [4] combined 2D image data and 3D model data redundancy space to generate complete OBJ file carriers, using multiple methods to protect the hidden information embedded in different types of carriers. This addressed the limitation of single carrier-type hiding algorithms that cannot "backup" critical information. Zaid et al. [5] used an irregular wavelet analysis method, embedding information into an appropriate resolution level by quantizing the norm of wavelet coefficient vectors, which provides a high data embedding rate. Ren Shuai et al. [6] proposed a multi-carrier information hiding algorithm for 3D model vertex partitioning, achieved by preprocessing 3D models that underwent geometric transformations to obtain a multi-carrier fusion state. In 2017, Hamidi et al. [7] proposed a blind robust watermarking algorithm for 3D semi-regular meshes, using edge norms as synchronization primitives. They embedded different information by modifying the norm of wavelet coefficient vectors associated with the lowest resolution level. In 2019, Hamidi et al. [8] utilized mesh significance of 3D semi-regular grids and quantized index modulation of wavelet coefficients to embed watermark information, achieving better invisibility and enhancing the algorithm's robustness against various attacks. In 2018, Liao et al. [9] focused on the distribution of embedded payloads in multi-image steganography. They established a framework for embedding data across multiple images and proposed two embedding strategies based on image texture complexity and distortion distribution, which achieved good statistical undetectability in multi-image steganography. In 2020, Liao et al. [10] further improved their algorithm by adaptively embedding secret information into the carrier. Based on the texture features of the images, the secret information was adaptively embedded into multiple images, effectively enhancing the algorithm's security performance. Yang et al. [11] proposed a multi-image information hiding algorithm that integrates multiple features of images. The algorithm iteratively selects the maximum effective capacity of each carrier image based on the complexity, quantity, and the number and size of the secret images, allowing the secret information to be distributed across multiple images with better security. Reversible data hiding schemes have also been extensively studied; for a comprehensive review [12,13].The 3D vertex projection technique provides a stable geometric domain for information embedding, enabling high imperceptibility and robustness, which makes it well suited for steganographic applications. In summary, most existing 3D model-based information hiding algorithms rely on a single 3D model as the carrier. The limited number of vertices and meshes available for

embedding secret data results in insufficient embedding capacity. Single-model algorithms exhibit poor robustness against attacks such as translation, rotation, random noise, and simplification, which often lead to loss or leakage of hidden information. Furthermore, due to the susceptibility of 3D models to various geometric attacks during transmission, current information hiding technologies struggle to ensure the security and integrity of secret information transmission.

In this work, the term multi-carrier refers to the use of multiple 3D models as concurrent carriers to overcome the embedding capacity and robustness limitations inherent in single-carrier schemes. Unlike traditional approaches that rely on a single model to embed all secret information, the proposed algorithm constructs a fusion carrier by geometrically aligning multiple 3D models through centroid coincidence and bounding sphere normalization. This multi-carrier strategy not only expands the available embedding region, thereby enabling higher-capacity information hiding, but also introduces spatial redundancy, which significantly enhances robustness against localized attacks. As a result, the embedded data is distributed across several carriers, making the system more resilient to partial model loss or deformation. Subsequent sections provide a detailed description of the multi-carrier fusion process and demonstrate its impact on the algorithm's performance through experimental validation.

Although a variety of 3D model-based information hiding algorithms have been developed, most existing approaches suffer from limited embedding capacity and poor robustness against geometric attacks, especially when using single-carrier models. These limitations restrict their applicability in real-world scenarios where data integrity and confidentiality under complex transmission environments are critical.

To address these challenges, this paper is motivated by the need to enhance both the robustness and embedding capacity of 3D steganographic systems. The objective of this work is to develop a multi-carrier information hiding algorithm that leverages the geometrical fusion of multiple 3D models and vertex neighborhood projection, allowing for reliable and covert communication under various attack scenarios, including translation, rotation, noise, simplification, and shear.

In response to the limitations of 3D model information hiding for a single carrier, this paper proposes a multi-carrier hiding algorithm based on vertex projection of 3D models. By integrating models, preprocessing, selecting candidate vertices, and performing vertex projection, the algorithm enables multi-carrier embedding, thereby enhancing its robustness.

The main contributions of this paper are summarized as follows:

A multi-carrier information hiding framework is proposed, which fuses multiple 3D models through centroid alignment and geometric normalization. This fusion overcomes the capacity limitations and improves the redundancy of traditional single-carrier schemes.

A vertex projection embedding method is designed, in which the secret information is hidden by adjusting the central vertex position in the 1-ring neighborhood projection plane. This approach enhances the algorithm's resistance to translation, rotation, and noise attacks.

A candidate vertex selection strategy based on vertex importance and 1-ring neighborhood area is introduced to balance invisibility and robustness, ensuring that embedding occurs in stable regions of the mesh.

Extensive experiments on benchmark 3D models demonstrate that the proposed algorithm achieves superior robustness and invisibility, with SNR values up to 47.87 dB and low BER under various attack scenarios, outperforming several state-of-the-art methods.

## 2. Key steps of information hiding algorithm

The proposed steganography algorithm in this paper consists of four main steps: multi-carrier fusion state combination, model preprocessing, vertex simplification and region selection, and secret information embedding rules.

The first step involves processing multiple 3D models with bounding spheres and using the model with the largest radius as the central model. The other 3D models are proportionally scaled to match the size of the central model and are fused at the centroid, forming the multi-carrier fusion state; The second step is to preprocess the multi-carrier fusion state to make it resistant to translation and rotation attacks; The third step involves selecting the vertices for embedding the

secret information based on vertex importance and the area of the one-ring neighborhood of the vertices, forming a set of candidate points; The fourth step projects the one-ring neighborhood of the vertices in the candidate point set onto their cross-sectional planes and embeds the secret information into the vertex data in the projection plane.

## 2.1. Combination method of multi-carrier fusion state

To construct a low-density 3D model information hiding algorithm and improve its robustness by increasing the carriers for embedding secret information, it is necessary to fuse multiple 3D models. In this paper, various multi-carrier fusion state algorithms are proposed based on different combination methods. The multi-carrier fusion state algorithms proposed in this paper are divided into preprocessing of individual carriers and combination methods. The specific algorithms are as follows:

Step 1: Perform bounding sphere for the 3D models. For $n$ 3D models, apply bounding sphere processing. According to formula (1), find the centroid $o_i$ $(x_{ic}, y_{ic}, z_{ic})$ of the 3D model as the model origin. $i$ represents the $i$-th 3D model, $1 \leq i \leq n$, $h_i$ denote the total number of vertices in the $i$-th 3D model. Using formula (2), calculate the radial distance $d_{ij}$ from each vertex to the centroid, and let $p_{ik}$ $(1 \leq k \leq j_i)$ be the farthest point from the centroid. $j_i$ represents the number of vertices in each 3D model. The bounding sphere is constructed with the distance $p_{ik}(x_{ik}, y_{ik}, z_{ik})$ between point $r_{ik}$ and the centroid as the radius, as shown in Fig 1. The bounding sphere of a 3D model is calculated. For $n$ 3D models, the bounding sphere processing is performed, and the centroid calculation formula for a 3D model is as follows:

$$x_{ic} = \frac{1}{h_i} \sum_{j=1}^{h_j} x_j, y_{ic} = \frac{1}{h_i} \sum_{j=1}^{h_j} y_j, z_{ic} = \frac{1}{h_i} \sum_{j=1}^{h_j} z_j$$

(1)

Where $o_\iota$ $(x_{\iota c}, y_{\iota c}, z_{\iota c})$ is the origin of the model, $i$ represents the $i$-th 3D model, $1 \leq \iota \leq n$, $h_i$ represent the total number of vertices of the $i$-th 3D model.

After calculating the centroid, the radial distance $d_{ij}$ from each vertex to the centroid needs to be computed, as follows:

(a) Bounding sphere diagram

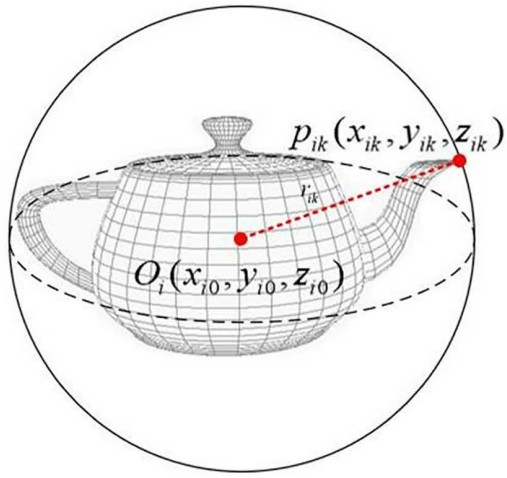

(b) Model bounding sphere diagram

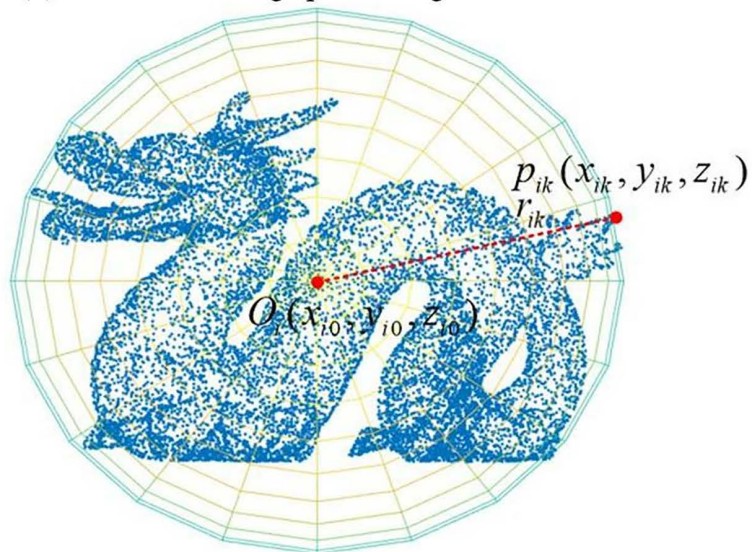

**Fig 1. Bounding Sphere Diagram of 3D Model.**

$$d_i = \sqrt{(x_i - x_0)^2 + (y_i - y_0)^2 + (z_i - z_0)^2}$$

(2)

Where $p_{ik}(1 \leq k \leq j_i)$ is the farthest point from the centroid, and $j_i$ is the number of vertices in each 3D model. The bounding sphere for the 3D model is processed using the radius $r_{ik}$, which is the distance between point $p_{ik}(x_{ik}, y_{ik}, z_{ik})$ and the centroid.

Step 2: Preprocessing of multiple models. Arrange the bounding sphere radii $r_{ik}$ of the $n$ 3D models. Based on the radius sizes, select the 3D model with the largest radius as the central model, denoted as $C$, with its radius as $R$. The bounding spheres of the other $n$-1 models are then scaled proportionally to match the radius $R$ of the central model's bounding sphere.

Step 3: Fusion of multiple models. Using centroid $o_i(x_{ic}, y_{ic}, z_{ic})$ as the overlap point, fuse the other $n$-1 3D models (processed with bounding spheres) with the central 3D model, ensuring that all vertices are evenly distributed across the entire coordinate system. The fused multi-carrier 3D model is shown in Fig 2.

## 2.2. Model preprocessing

To enhance the robustness of the 3D model against geometric attacks during transmission, preprocessing is required to ensure geometric transformation invariance. The specific processing steps are as follows.The processing flow is shown in Fig 3.

Step 1: Redefine the coordinate system. Let $U$ be the set of vectors between all vertices of the 3D model and the centroid. Calculate the vector $u_1 : |u_1| = \max\{|u| : u \in U\}$ with the largest magnitude among them. Next, find the vertex $V_d$ that is farthest from $u_1$ and its projection $O_N$ onto $u_1$. Meanwhile, define vector $u_2$ as the one formed by vectors $V_d$ and $O_N$. Determine $O_N$ as the origin of the new coordinate system, $u_2$ as the $x$-axis, and $u_1$ as the $z$-axis of the new coordinate system. The two endpoints of vector $u_2$ and the coordinate origin $O_N$ are not involved in the embedding process.

Step 2: Rotation and Translation. Perform translation and rotation on the 3D model to transform it into the new coordinate system. This involves setting $O_N$ as the new origin of the 3D model's coordinate system, aligning vector $u_1$ with the $x$-axis, and aligning vector $u_2$ with the $z$-axis.

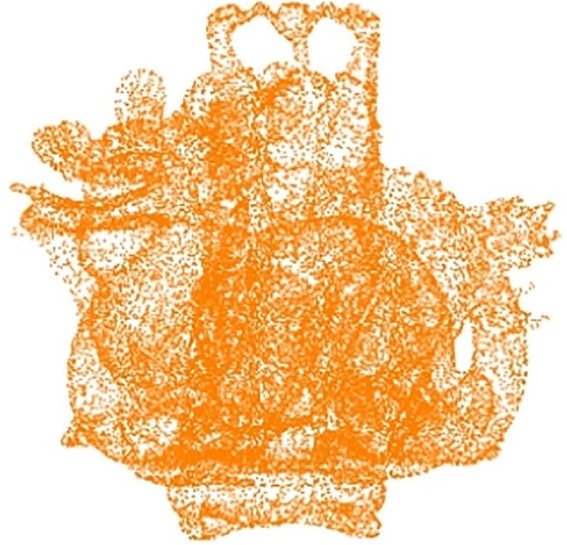

**Fig 2. Fusion state point cloud model of 3D models.**

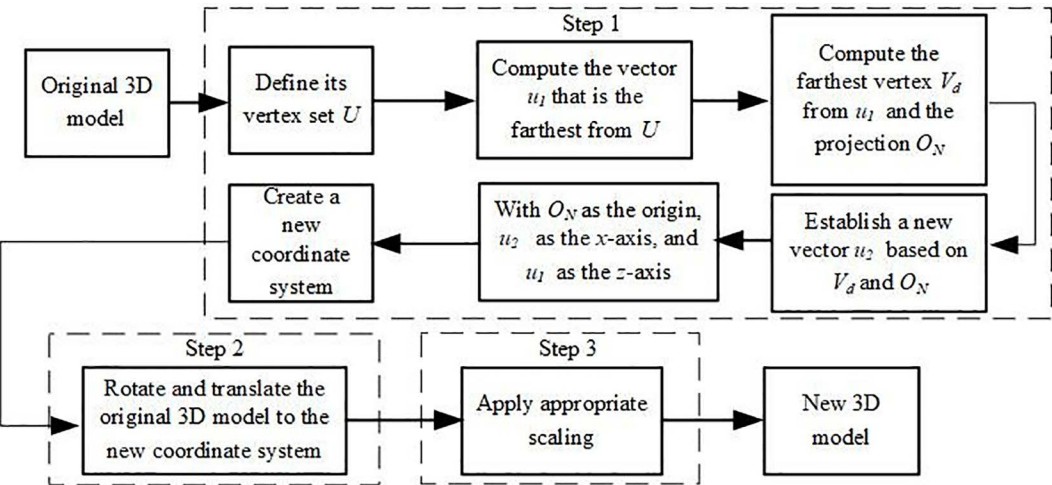

**Fig 3. Model preprocessing flowchart.**

Step 3: Uniform Scaling. Perform a scaling transformation on the 3D model using $|O_N V_d|$ as the scaling factor.

Through the transformations in the above steps, a 3D model with geometric invariance can be obtained, represented as $M_T = \{V_T, E_O\}$, with $V_T$ as the set of vertices, $V_T = \{v_i | v_i = (x_i, y_i, z_i), i = 1, 2, 3, ..., n\}$.

## 2.3. Vertex simplification and region selection

As the resolution of 3D models increases, the data volume of vertices and triangular facets in the models becomes larger. To improve the algorithm's processing speed and reduce storage space, it is necessary to simplify 3D models with large amounts of data. 3D model simplification refers to reducing the number of vertices or triangular facets that make up the 3D model while approximating the original model's shape as closely as possible. This involves removing unnecessary vertex or facet information from the model, and the degree of simplification is adjusted based on the amount of secret information to be embedded.

The importance of a vertex in a 3D model is defined as the distance from the vertex to the average plane formed by the 1-ring neighborhood points of that vertex, denoted as $d(p, f)$ [14–17]. The 1-ring neighborhood points of a vertex refer to the points that are directly adjacent to it. The average plane refers to the plane constructed by the normal vector $\vec{n}_i$ of the triangular mesh of the vertex's 1-ring neighborhood and the center coordinates $\vec{x}_i$ of the average plane. The formula for calculating the vertex importance is shown in Equation (3).

$$I(p) = d(p, f) = \left| \vec{n} \cdot (\vec{p} - \vec{x}) \right|$$
$$\vec{N} = \frac{\sum \vec{n}_i A_i}{\sum A_i}, \quad \vec{n} = \frac{\vec{N}}{|\vec{N}|}, \quad \vec{x} = \frac{\sum \vec{x}_i A_i}{\sum A_i}$$

(3)

Where $\vec{p}$ is the normal vector of the center coordinates, and $A_i$ represents the area of the triangular facets in the 1-ring neighborhood of the vertex.

For example, the importance of vertex $p$ is shown in Fig 4.

The difference between the vertices of a simplified 3D model and the pixels of a 2D image is that vertices do not have an implicit order. In a 3D model containing secret information, it is difficult to extract secret data based on the 3D coordinates and the embedding locations. Therefore, we choose to determine the candidate vertex set based on the geometric properties of the 3D model, specifically the area of the vertex's 1-ring neighborhood. If the area of a vertex's 1-ring

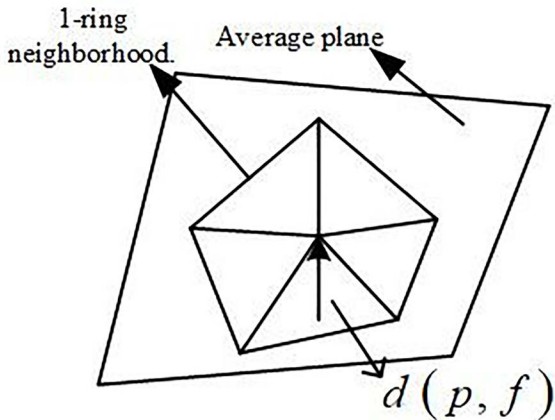

**Fig 4. Schematic of the average plane and vertex importance.**

neighborhood is large, it is more vulnerable to mesh subdivision attacks. However, regions with smaller 1-ring neighborhood areas are more susceptible to mesh simplification attacks. Thus, we select vertices with a medium-sized 1-ring neighborhood area for embedding secret information in their neighborhood, ensuring the robustness of the hidden data. The vertices' 1-ring neighborhood areas are sorted from smallest to largest $(K_1,..., K_j)$, and the vertex with the median 1-ring neighborhood area is denoted as $K_c$. If the size of the embedded secret information is $M$, then the set of vertices with medium 1-ring neighborhood areas is $(K_c - M/2, K_c + M/2)$. The formula for calculating the area of the 1-ring neighborhood of a vertex is shown in Equation (4).

$$S_i = \frac{1}{J} \sum_{j=1}^{J} s_j$$

(4)

Where $J$ is the total number of vertices in the 1-ring neighborhood of $v_i$, and $s_j$ is the area of the triangular mesh in the 1-ring neighborhood of $v_i$.

After determining the candidate vertex set, the vertices in the set are sorted based on the area of their 1-ring neighborhoods. If two vertices have the same 1-ring neighborhood area, they are further sorted by their 3D coordinates. The coordinates are compared sequentially, starting with the $x, y, z$ coordinate; the vertex with the smaller $x, y, z$ coordinate is placed first. Then, the secret information is embedded according to the order of the vertices in the list. For example, if vertex $v_i$ is in the first position in the candidate set, the first bit of the secret information will be constructed by extracting geometric features from the 1-ring neighborhood of $v_i$. The geometric features of the 1-ring neighborhood of the second vertex in the candidate set will represent the second bit of the secret information.

## 2.4. Secret information embedding rules

The information hiding algorithm in this paper is divided into the following steps, and the flow of the information hiding algorithm is shown in Fig 5.

Step 1: Secret Information Processing. The secret information is Huffman encoded, and the new information sequence obtained after encryption is hidden. The key is the Huffman encoding table specifically designed for the secret information. The original 3D model $M_1, M_2, \cdots, M_n$ is read, and the centroid of each model is determined based on the definition. For each 3D model, a bounding sphere is generated to provide a base model for constructing the fusion state.

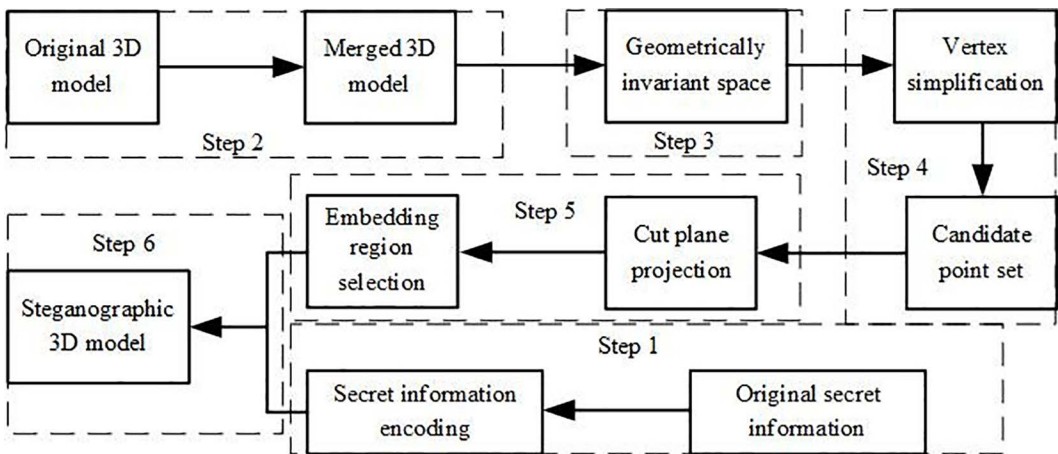

**Fig 5. Flowchart of Secret Information Hiding.**

Step 2: Constructing the Multi-Carrier Fusion State. Based on the composition method of the multi-carrier fusion state outlined in Section 2.1, the center model and overlap points are selected. Multiple 3D models are combined such that all vertices are evenly distributed across the entire coordinate system, providing an environment for embedding large-capacity secret information.

Step 3: Model Preprocessing. The model composed of multiple carriers undergoes preprocessing. According to the description in Section 2.2, the 3D model is transformed into an affine-invariant space, and a new coordinate system is constructed based on principal component analysis (PCA) to resist operations such as uniform scaling, rotation, and translation.

Step 4: Vertex Simplification and Region Selection. According to the vertex importance formula (3) defined in Section 2.3, the importance of all vertices in the multi-carrier fusion state is calculated. The results are sorted in descending order, and the less important vertices are removed from the model to improve the algorithm's processing speed. A candidate vertex set is determined based on the area of the 1-ring neighborhood of the simplified vertices. The area of the 1-ring neighborhood of each vertex is calculated and sorted, and secret information is embedded in the vertices in descending order of their neighborhood areas. If vertex $v_i$ ranks first, its 1-ring neighborhood vertices will embed the first bit of secret information. Additionally, the neighborhood information of vertices with medium 1-ring neighborhood areas is used to embed secret information, avoiding grid simplification attacks on smaller areas and grid subdivision attacks on larger areas, thus enhancing the robustness of the secret information. After vertex simplification and region selection, the final candidate point set $I$ for embedding secret information is obtained.

Step 5: Tangential Plane Projection.The vertex and its 1-ring neighborhood form a unique surface, so the tangent plane of the vertex's 1-ring neighborhood is also unique [18], determined by the vertex's coordinates and the normal direction of its 1-ring neighborhood, each vertex in the candidate set is projected onto its own tangential plane. The origin and axes of the projection image align with the coordinate axes of the tangential plane. Encrypted information is embedded in the projection image of the vertex, improving the invisibility of the algorithm.

Step 6: Secret Information Embedding. After projection, the vertices are represented by 2D coordinates in the projection image. Secret information is embedded by altering the position of the central vertex in the projection image, changing the distances between it and its 1-ring neighborhood vertices. Let the central vertex of the 1-ring neighborhood be $v_i(x_i, y_i)$, the vertex with the smallest $y$-axis coordinate in the projection image be $v_m(x_m, y_m)$, and the vertex with the largest $y$-axis coordinate be $v_M(x_M, y_M)$. If the secret information to be embedded is 0, set $y_i = 1/2 y_m$, and if the secret information to be

embedded is 1, set $y_i = 1/2y_M$. Correspondingly, adjust the position of the vertex in 3D space, ensuring that $v_i$ remains within the 1-ring neighborhood vertices and that the 1-ring neighborhood area is not changed.

Step 7: Split the multi-carrier 3D model fusion state containing secret information, ultimately obtaining multiple 3D models $M_1{}', M_2{}', ......, M_n{}'$, each embedded with secret information.

## 2.5. The complete steps of secret information extraction

The process of secret information extraction in the algorithm proposed in this paper is similar to the embedding process. The information extraction flow is shown in Fig 6.

Step 1: Reconstruct the carrier $M_1{}', M_2{}', ......, M_n{}'$ containing secret information according to the combination steps of the multi-carrier fusion state $S_1$ to obtain the 3D model multi-carrier fusion state containing secret information.

Step 2: After transforming the 3D model multi-carrier fusion state into a geometrically invariant space, perform vertex simplification based on the importance of the vertices. Select vertices with higher importance values to embed the secret information, thereby improving the algorithm's processing speed.

Step 3: Determine the set of secret-containing vertices. Based on the area of the 1-ring neighborhood of the 3D model, identify the candidate vertex set. Calculate the coordinates, angles, and 1-ring neighborhood area of the simplified vertices, and sort them in descending order. The secret information is then extracted sequentially according to this order.

Step 4: Determine the tangent plane of the 1-ring neighborhood of the vertex and project its 1-ring neighborhood to generate a projection image. In the projection image, extract the secret information embedded in the model vertices according to formula (5).

$$w_n = \begin{cases} 0, y_i = \frac{1}{2}y_m \\ 1, y_i = \frac{1}{2}y_M \end{cases}$$

(5)

Where $y_i$ is the y-axis coordinate value of a vertex in the projection image of its tangent plane from the candidate set, and $y_m, y_M$ represents the minimum and maximum y-axis coordinate values in the 1-ring neighborhood projection image. If $y_i = 1/2y_m$, the embedded secret information is 0; if $y_i = 1/2y_M$, the embedded secret information is 1.

Step 5: After fusion, transformation, computation, and sorting of the secret-containing 3D model, the bitstream can be extracted in sequence. This bitstream represents the encoded secret information. When combined with the encoded original secret information, the bit error rate of the algorithm on the model can be calculated.

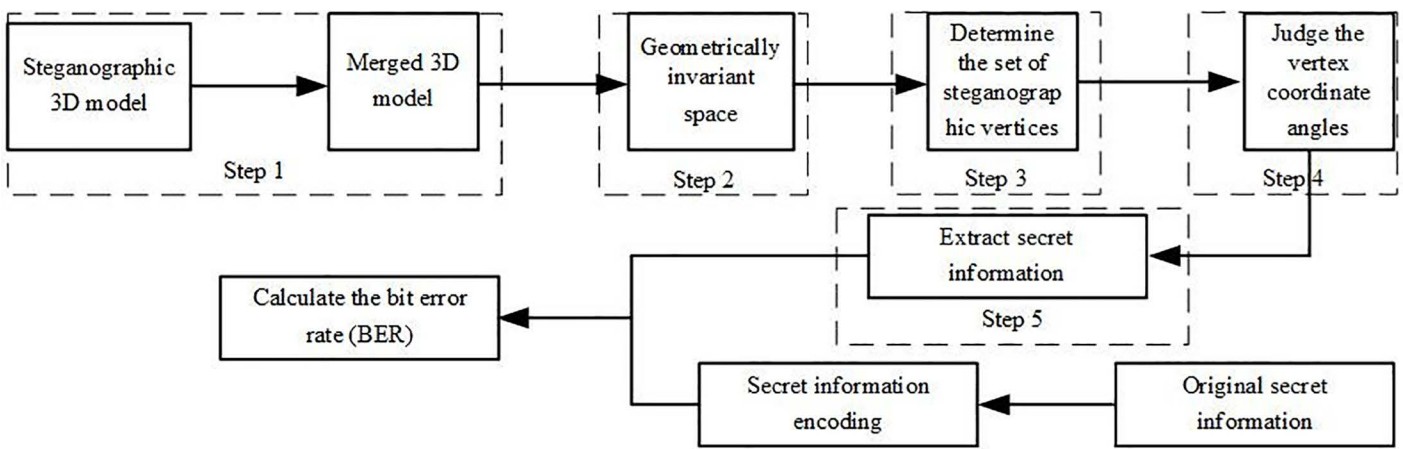

**Fig 6. Flowchart of secret information extraction.**

## 3. Simulation experiments and comparative analysis

During transmission, the secret-containing carrier may be subjected to various types or intensities of attacks, and the hidden secret information could be modified or lost. In this paper, experiments and simulations of the algorithm are conducted in the Autodesk 3Ds Max 2016, MatlabR2018a, and Meshlab environments. Three robust information hiding algorithms are chosen for comparison, including the Zero Watermark Rotation Attack (ZWRA) method proposed in [18] to counter rotation attacks, the Triangular Strips Progressive Grid (TSPD) algorithm based on progressive grids and triangle strips proposed in [19], and the Spatial Subdivision and Spatial Coding (SSC) algorithm for 3D data hiding proposed in [20,21]. The selected test models are from the Stanford 3D Scanning Repository and the Princeton Shape Benchmark, including 3D models: bunny, dragon, happy_vrip, and xyzrgb_dragon. The original models of the 3D objects are shown in Fig 7, with data information listed in Table 1, which includes the number of vertices and the number of triangular faces. The tested attacks include translation, rotation, simplification, random noise (Gaussian noise), and shear attacks.

Regarding the complexity of the algorithm proposed in this paper, Matlab experiments show that the time complexity is determined by the maximum complexity of each fixed process in the embedding and extraction procedure. These processes include calculations for the fusion of multiple 3D models, geometric coordinate transformations of individual models, and embedding coordinate calculations. Among these, the most complex calculations are the coordinate transformation and vertex importance calculation, which require traversing all vertices. Therefore, the time complexity is related to the total number of vertices in the carrier model, denoted as N, and is O(N). The space complexity is determined by the storage space needed to store all the vertex data, which is related to the model.

### 3.1. Invisibility experiment

In this part of the experiment, since the degree of modification of the model can be quantitatively calculated using certain computational methods, the invisibility is measured here based on the Signal-to-Noise Ratio (SNR) algorithm. The SNR

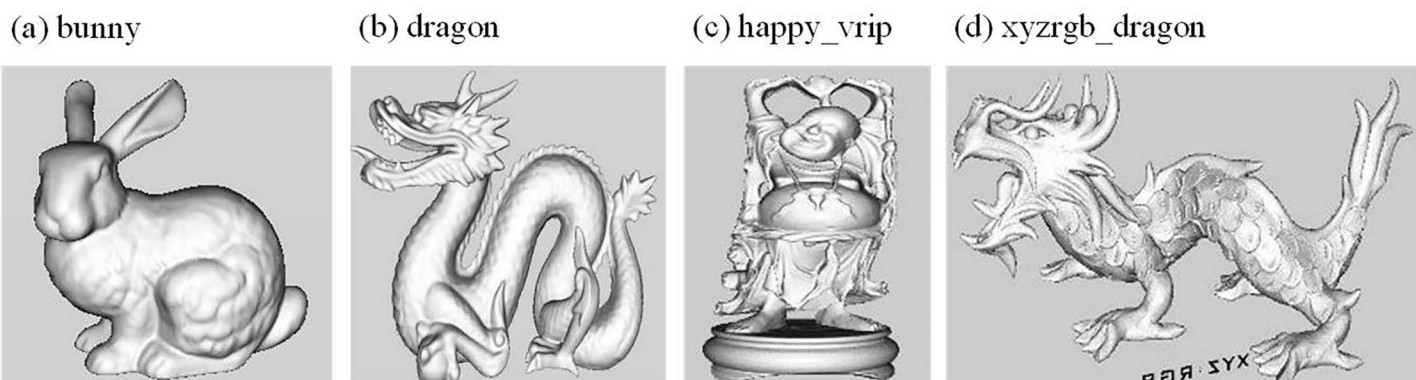

**Fig 7. Original 3D Model.**

**Table 1. 3D model data information.**

| Model | Number of vertices | Number of triangular faces |
|---|---|---|
| bunny | 35947 | 68451 |
| dragon | 437645 | 871414 |
| happy_vrip | 543652 | 1067716 |
| xyzrgb_dragon | 3609600 | 7219045 |

represents the objective distortion between the model with embedded secret information and the original model before embedding the secret information. The calculation method is shown in equation (6). A smaller SNR value indicates a larger amount of effective noise, meaning the test model has significant visual differences, and the algorithm's invisibility is poor. Conversely, a larger SNR value indicates better invisibility of the algorithm.

$$SNR = 10log_{10}\left[\frac{\sum_{i=0}^{n}\left(x_i^2 + y_i^2 + z_i^2\right)}{\sum_{i=0}^{n-1}\left((x_i' - x_i)^2 + (y_i' - y_i)^2 + (y_i' - y_i)^2\right)}\right]$$

(6)

Where $n$ is the number of vertices in the model, $(x_i, y_i, z_i)$ represents the vertex coordinates of the original model, and $(x_i', y_i', z_i')$ represents the vertex coordinates of the model with embedded secret information.

The 3D model with embedded secret information is shown in Fig 8. The changes in the 3D point cloud model before and after embedding the secret information are minimal. Compared to the original model, the model with embedded information is visually difficult to distinguish from the original, indicating that the algorithm has good invisibility.

To validate the invisibility of the algorithm, a comparative experiment was conducted with the algorithms in references [18–20]. The SNR values of the four algorithms are shown in Table 2. As seen in Table 2, the proposed algorithm significantly improves the SNR compared to the other algorithms. This is because the algorithm selects vertices with moderate 1-ring neighborhood areas as carriers, projects the 1-ring neighborhood of the vertices onto their tangent planes, and embeds secret information into the projection images. This results in minimal impact on the position of the vertices in 3D space. Additionally, the secret information is embedded into a point cloud model with multiple carriers in a fused state, effectively expanding the embedding region. The hiding capacity can be adjusted by altering the number of carriers based

**(a) bunny**　　　　　**(b) dragon**　　　　　**(c) happy_vrip**　　　　　**(d) xyzrgb_dragon**

**Fig 8. The model with embedded secret information.**

**Table 2. Signal-to-noise ratio of the 3D model.**

| Method | 3D model | | | |
|---|---|---|---|---|
| | dragon | bunny | happy_vrip | xyzrgb_dragon |
| **Hou G, et al. [18]** | 36.37 | 38.25 | 42.91 | 44.79 |
| **Gao K,et al. [19]** | 38.83 | 39.47 | 44.53 | 45.32 |
| **Gao K,et al. [20]** | 40.78 | 42.35 | 45.72 | 46.58 |
| **The algorithm proposed in this paper** | 41.28 | 44.76 | 46.56 | 47.87 |

on the size of the secret information, making it suitable for large-capacity applications. Since the secret information is distributed across multiple models, the changes to each individual model are minimal, which contributes to the algorithm's good invisibility.

As shown in Table 2, the proposed algorithm achieves the highest SNR values across all tested models, indicating minimal distortion and strong invisibility compared to the baseline methods.

In addition, the algorithm's resistance to analysis can be demonstrated using the invisibility experimental results. This is because current steganalysis algorithms based on local feature sets and Laplacian smoothing suffer from several issues, such as limited sample scenarios, insufficient accuracy of the analyzer, and the ability to only analyze single-model, single-steganography algorithms. Therefore, further research is needed to develop steganalysis methods for multi-3D model fused states. From the perspective of steganalysis itself, the steganalysis tool measures the degree of modification of the 3D model before and after embedding the secret information. Invisibility currently reflects the algorithm's resistance to analysis.

### 3.2. Robustness experiment

The following robustness experiments use the 3D models from Table 1 and compare the TSPD algorithm, ZWRA algorithm, and SSC algorithm with the multi-carrier fusion algorithm proposed in this paper. The experiments employ the same secret information embedding, resulting in 16 types of secret-embedded carriers [22]. Since it is crucial to extract the secret information intact after the model is encrypted and subjected to complex environments, the algorithm evaluates the robustness of different methods using the Bit Error Rate (BER). BER represents the ratio of the number of incorrect bits in the extracted secret information to the total number of information bits, with a value range of [0,1]. The closer the BER value is to 0, the fewer the number of incorrect bits, indicating stronger robustness of the model. The calculation method is shown in Equation (7).

$$BER = \frac{\sum_{i=1}^{N} (w'_i \neq w_i)}{N}$$

(7)

Where $w_i$ and $w_i'$ represent the original secret information sequence and the extracted secret information sequence at the $i$-th position, respectively, and $N$ is the number of bits in the original secret information sequence.

Since the secret information is embedded into the neighborhood of vertices selected from multiple 3D models, the experimental data will differ when any single 3D model is attacked compared to when multiple 3D models are attacked simultaneously. Therefore, the final algorithm results are taken as the average values under different levels of attacks.

The BER values of the secret-bearing carrier under translation attacks of different lengths are shown in Fig 9, and the BER values under rotation attacks at different angles are shown in Fig 10. As shown in the figures, the proposed algorithm can effectively resist translation and rotation attacks. This is because the algorithm selects the geometric feature of the 3D model, specifically the 1-ring neighborhood area of the vertices, as the carrier for embedding secret information. Regardless of whether the attacker applies translation of any length or rotation at any angle to the model, the 1-ring neighborhood area of the vertices remains unchanged [23]. Therefore, the correct location of the embedded secret information can still be determined by the 1-ring neighborhood of the vertices [24,25]. Moreover, the secret information is embedded in the relative positions between the central vertex and other vertices. Translation and rotation attacks affect the coordinates of all vertices simultaneously, altering their spatial positions, but they do not affect the relative positions between vertices. This allows the algorithm to correctly determine whether the embedded secret information is 0 or 1 and successfully extract the secret information. Therefore, compared to the other three algorithms, the proposed algorithm exhibits a lower BER value under different levels of attack, demonstrating a stronger ability to resist translation and rotation attacks [26,15].

The BER remains consistently low under both translation and rotation, validating the geometric invariance of the projection-based embedding scheme.

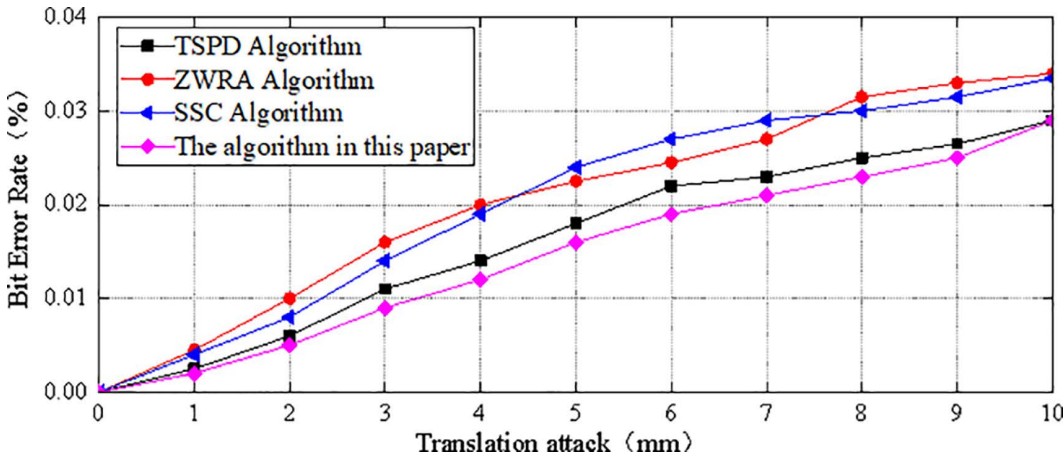

**Fig 9. The BER value after translation attack.**

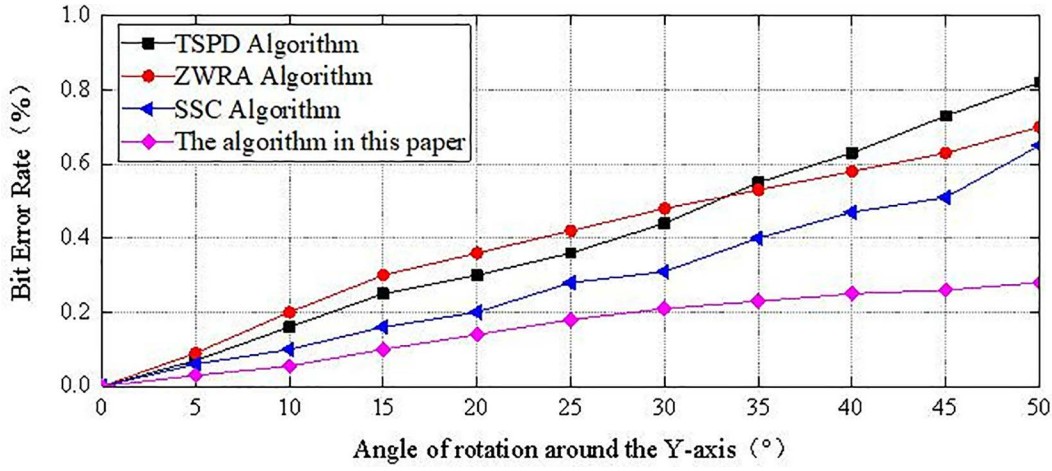

**Fig 10. The BER value after rotation attack.**

Fig 11 shows the robustness of four methods against simplification attacks when the vertex simplification rate ranges from 0% to 30%. The main target of vertex simplification attacks is vertices with lower importance or smaller areas, so that the attacked secret-bearing carrier remains undetected by the receiver, allowing the attacker to mislead the receiver into obtaining incorrect secret information. In the proposed algorithm, the vertices selected for embedding secret information are those with higher importance and a moderate 1-ring neighborhood area. Even when simplification attacks cause a small number of vertices to be lost, the algorithm is still able to extract the correct secret information to a large extent. Under the same level of simplification attack, the results of the proposed algorithm outperform the other three methods.

Even under 30% vertex simplification, the BER of the proposed method remains significantly lower than other algorithms, demonstrating strong robustness.

Fig 12 shows the secret-bearing model after being subjected to noise attacks of different intensities, and Fig 13 illustrates the robustness of the four methods against noise when the noise intensity $s$ increases from 0 to 0.001. As shown in the figures, when the added random noise intensity is relatively small, i.e., less than $0.2 \times 10^{-3}$, all four methods exhibit

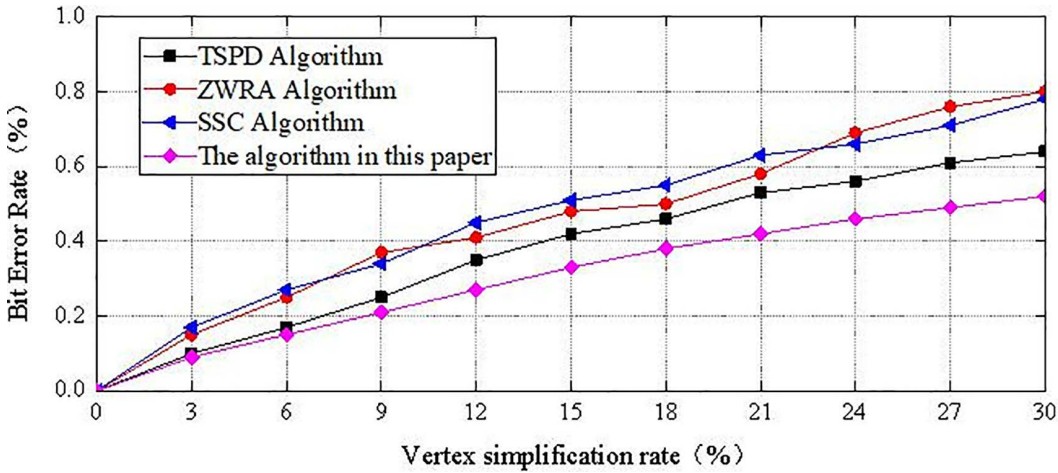

**Fig 11. The BER value after simplification attack.**

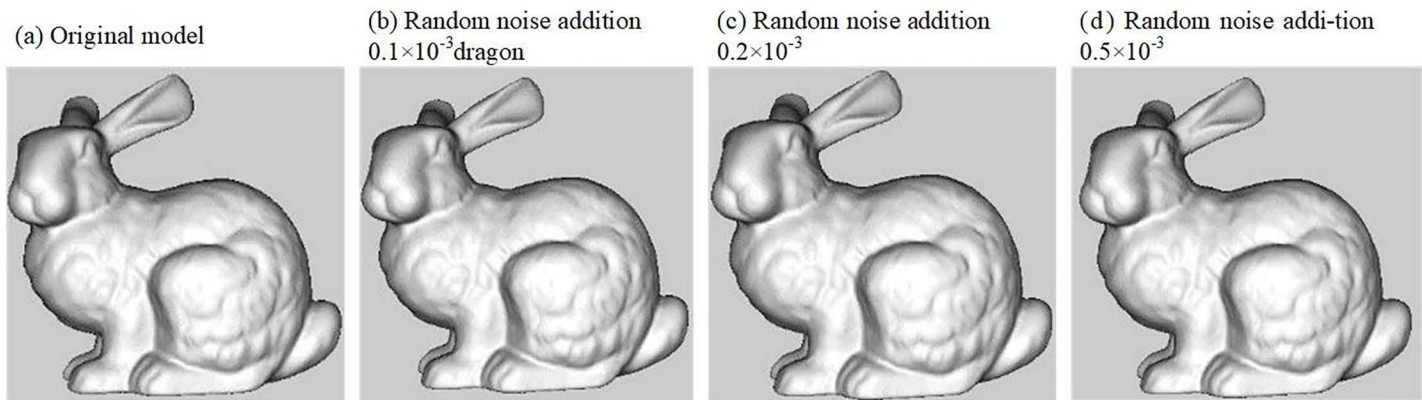

(a) Original model

(b) Random noise addition $0.1 \times 10^{-3}$ dragon

(c) Random noise addition $0.2 \times 10^{-3}$

(d) Random noise addi-tion $0.5 \times 10^{-3}$

**Fig 12. Noise attack.**

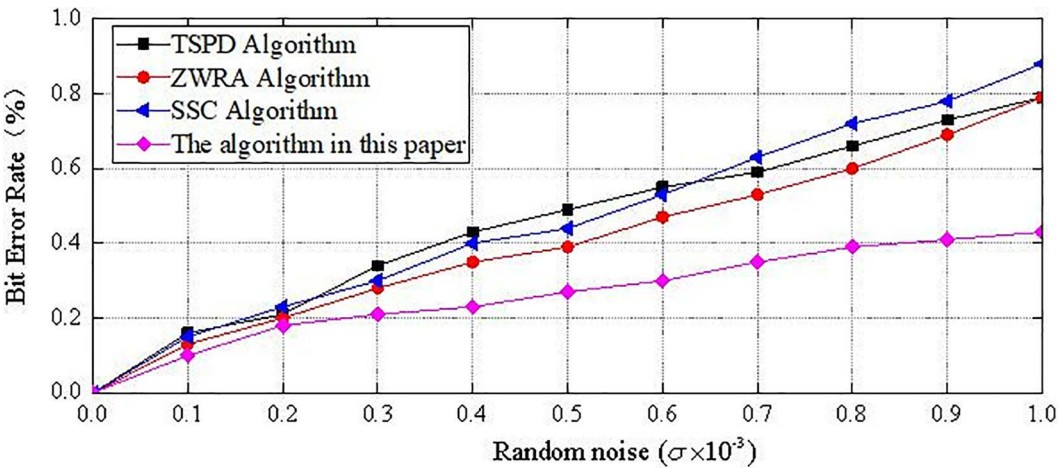

**Fig 13. The BER value after noise attack.**

better robustness. However, as the noise intensity increases, the proposed algorithm performs significantly better in resisting noise attacks compared to the other methods. This is because noise attacks disturb the vertex coordinates, which interferes with the extraction of secret information, but they do not affect the 1-ring neighborhood area of the vertices or the relative positions between vertices. Therefore, even under high-intensity noise attacks, the proposed method is still able to successfully extract the secret information.

The BER increases more slowly under noise attacks in the proposed algorithm, suggesting its effectiveness in resisting vertex perturbation.

As shown in Fig 14, when the model is subjected to shear attacks, for the multi-carrier fusion state, when the vertex shear rate increases from 0% to 30%, Fig 15 shows the performance of the four methods against shear attacks. Compared to resisting simplification attacks, the algorithm's ability to resist shear attacks is weaker, with no significant advantage over the other three algorithms. This is because shear attacks involve partial destruction of the model, and the algorithm does not embed the secret information repeatedly across different vertices. As a result, when the model is heavily sheared, the secret information that has been cut cannot be extracted from the sheared model, leading to the loss of some secret information. Further improvements to address such attacks are needed in future algorithm research.

In addition to robustness metrics, preliminary observations suggest that the multi-carrier embedding strategy and the use of entropy-based encoding schemes (e.g., Huffman coding) offer a certain level of resilience against basic steganalysis and statistical cryptanalysis. Nonetheless, a comprehensive evaluation under advanced cryptanalytic models (e.g., chosen-message attacks, stego-only attacks) [27,28] is not yet conducted and will be considered in future research.

The proposed technique primarily considers geometric attacks such as translation and rotation, as well as topological attacks like mesh simplification, which are common and challenging scenarios in 3D steganography.

## 4. Conclusion

In this paper, we propose a multi-carrier information hiding algorithm based on vertex projection of 3D models to address the limitations of embedding capacity and robustness in single-carrier steganographic schemes. By fusing multiple 3D models into a centroid-aligned and geometrically invariant fusion carrier, and embedding secret information through projection-based vertex manipulation, the algorithm significantly enhances both invisibility and resistance to geometric attacks.

Experimental results demonstrate the superiority of the proposed method. In terms of invisibility, the algorithm achieves a peak signal-to-noise ratio (SNR) of 47.87 dB on the xyzrgb_dragon model, which outperforms existing methods such as the SSC (46.58 dB), TSPD (45.32 dB), and ZWRA (44.79 dB) algorithms. Regarding robustness, the proposed method

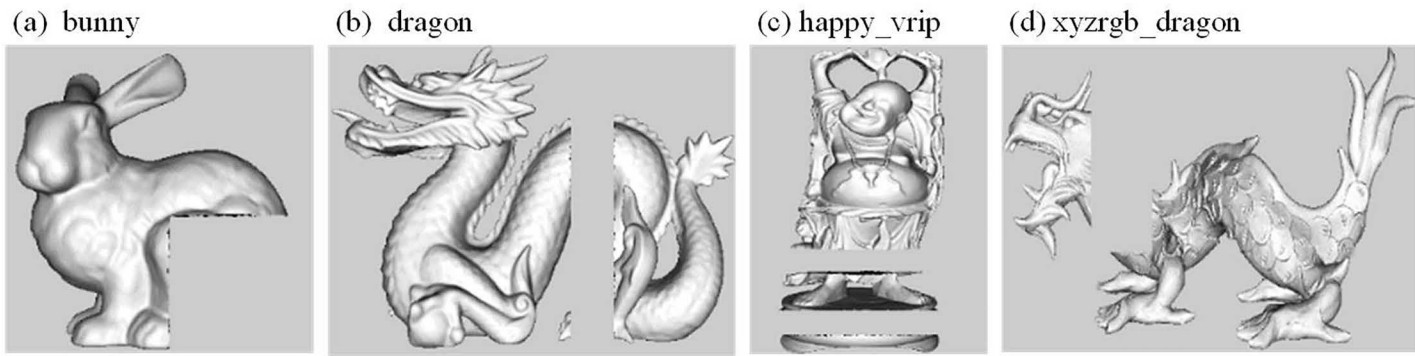

**Fig 14. Shear attack.**

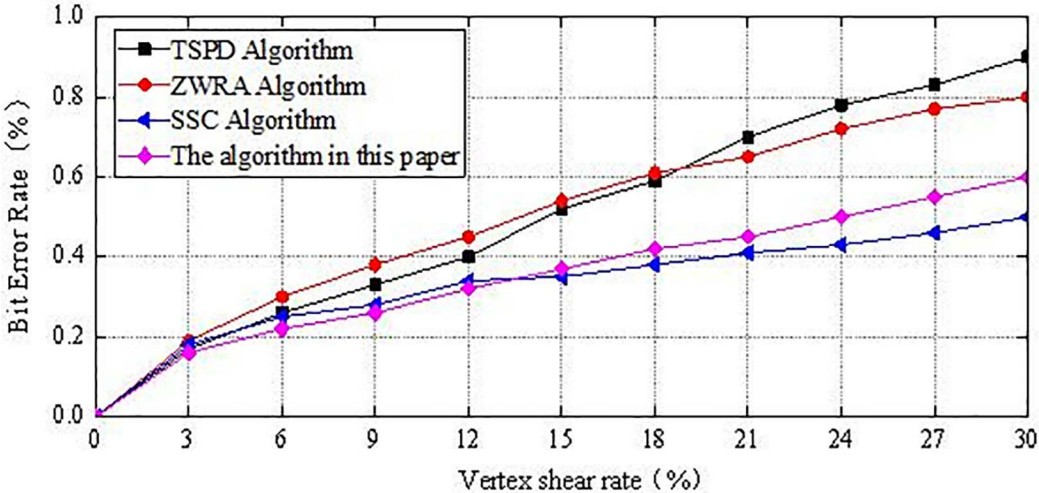

**Fig 15. The BER value after shear attack.**

maintains a bit error rate (BER) lower than 0.1 under 30% vertex simplification, and remains resilient under random noise levels up to 0.001, with BER performance consistently superior to competing algorithms.

Compared with existing methods such as SSC, TSPD, and ZWRA, the proposed algorithm improves the peak SNR by up to 1.3 dB, and reduces the BER under simplification and noise attacks by over 30%, demonstrating a significant enhancement in both invisibility and robustness.

Nevertheless, the algorithm has certain limitations. Specifically, while it performs well under translation, rotation, simplification, and noise attacks, its robustness against large-scale shear attacks is relatively weak. This is mainly due to the absence of redundant embedding or multi-location replication strategies, which results in partial information loss when critical vertices are sheared away. Moreover, the algorithm assumes precise alignment during multi-carrier fusion and relies on geometric integrity, which may not hold under extreme deformations or lossy transmission channels.

The integration of multiple carriers plays a pivotal role in improving the embedding capacity and attack resilience of the system, serving as a fundamental enhancement over conventional single-model steganographic techniques.

Future work will focus on enhancing the algorithm's resistance to destructive attacks such as large-scale shearing and affine distortions, potentially by introducing redundant encoding, adaptive region replication, or learning-based reconstruction to preserve information completeness and integrity under more adverse conditions.

## Supporting information

**S1 Data. Data.**

(RAR)

## Author contributions

**Conceptualization:** Yuefa Ou, Jie Ke, Chenglong Zhao.

**Data curation:** Yuefa Ou.

**Formal analysis:** Yuefa Ou, Zhuoyi Dan.

**Funding acquisition:** Chenglong Zhao.

**Investigation:** Mingkun Yang, Haobo Chen, Zhuoyi Dan.

**Methodology:** Yuefa Ou, Jie Ke, Mingkun Yang, Haobo Chen.

**Project administration:** Chenglong Zhao.

**Software:** Mingkun Yang, Haobo Chen, Zhuoyi Dan, Chenglong Zhao.

**Supervision:** Jie Ke, Mingkun Yang.

**Validation:** Haobo Chen, Zhuoyi Dan, Chenglong Zhao.

**Visualization:** Jie Ke.

**Writing – original draft:** Yuefa Ou.

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
