## [Decision Letter · Decision Letter 0]

17 Mar 2025

PONE-D-25-02638Multi-carrier Information Hiding Algorithm Based on Vertex Projection of 3D Model

PLOS ONE

Dear Dr. ZHAO,

Thank you for submitting your manuscript to PLOS ONE. After careful consideration, we feel that it has merit but does not fully meet PLOS ONE’s publication criteria as it currently stands. Therefore, we invite you to submit a revised version of the manuscript that addresses the points raised during the review process.

We look forward to receiving your revised manuscript.

Kind regards,

Giridhar Maji, Ph.D.

Academic Editor

PLOS ONE

 [National Natural Science Foundation of China (NSFC) Youth Project No. 61702050; 2024 Guangxi Uni-versities Enhancement of Research Capability for Young and Middle-aged Teachers Project No. 2024KY0447.]. 

5. We note that your Data Availability Statement is currently as follows: [All relevant data are within the manuscript and its Supporting Information files.]

Additional Editor Comments:

Serious concerns were raised by Reviewer#4. Proper justification and rigorous revision is expected from the authors.

Reviewers' comments:

Reviewer's Responses to Questions

**Comments to the Author**

1. Is the manuscript technically sound, and do the data support the conclusions?

Reviewer #1: Yes

Reviewer #2: Yes

Reviewer #3: Yes

Reviewer #4: No

2. Has the statistical analysis been performed appropriately and rigorously? 

Reviewer #1: Yes

Reviewer #2: Yes

Reviewer #3: Yes

Reviewer #4: No

3. Have the authors made all data underlying the findings in their manuscript fully available?

Reviewer #1: Yes

Reviewer #2: Yes

Reviewer #3: Yes

Reviewer #4: Yes

4. Is the manuscript presented in an intelligible fashion and written in standard English?

Reviewer #1: Yes

Reviewer #2: Yes

Reviewer #3: Yes

Reviewer #4: Yes

5. Review Comments to the Author

Reviewer #1: Comments:

1.The abstract should include some of the numerical results to reflect the superiority of the proposed steganography algorithm.

2.The mentioned references in the introduction section are not very recent. The authors are encouraged to enrich their introduction section with the following list of recent and directly related works:

a.https://ieeexplore.ieee.org/document/10643045

b.https://ieeexplore.ieee.org/document/10323523

c.https://ieeexplore.ieee.org/document/10163067

d.https://link.springer.com/article/10.1007/s11042-020-09437-w

e.https://link.springer.com/article/10.1007/s00530-023-01253-0?fromPaywallRec=false

3.Figure 1 should be provided at a higher resolution. Currently, it is hard to read the information depicted on it.

4.In the conclusions section:

a.Include some of the numerical results to reflect the superiority of the proposed algorithm.

b.What are the limitations of this algorithm

Finally, I would like to commend the authors on this wonderful work.

Reviewer #2: About the manuscript “Multi-carrier Information Hiding Algorithm Based on Vertex Projection of 3D Model”, the authors present a multi-carrier information hiding algorithm based on vertex projection of 3D models to improve the robustness of single 3D model as a carrier for information

Hiding. I think this theme is more interesting. Overall, it sounds technically feasible, and the structure of the manuscript is complete. Yet, here are some suggestions which should be taken into account to improve the current version.

1.Multi-carrier, but there doesn't seem to be much in the text.

2.From the references, most of them are relatively old, so it is suggested to briefly discuss them with some new work.

DOI :10.3390/math12243917; DOI:10.1016/j.eswa.2024.123190;

DOI: 10.1007/s11071-021-06206-8; 10.1007/s00371-023-02812-2。

3.The current experimental analysis is too simple, the research topic is Information Hiding Algorithm, and a richer experimental comparison is recommended according to the analysis method in literature 15.

4.For the information hiding method, whether its security can resist the attack of cryptanalysis can not be ignored, it is suggested to make a brief statement of the cryptanalysis work.

DOI: 10.1016/j.eswa.2024.123748

DOI: 10.1016/j.eswa.2023.121514;

5.The current article draft still contains a significant number of typographical and grammatical errors. Additionally, it is recommended to study the writing and expression of higher-level literature, and to carefully check and correct these issues.

Reviewer #3: Review Commends

Thank the authors for the submission of the manuscript entitled “Multi-carrier Information Hiding Algorithm Based on Vertex Projection of 3D Model ”. The paper needs more work to achieve a publishable level due to the following reasons.

1. Motivation and objective of the paper is missing.

2. What is the contribution of this paper?

3.Authors have not included some recent existing paper from year (2023 and 2025) in Introduction Part. So, Authors must cite some recent work related in the open

i.A cellular automata based secured reversible data hiding scheme for dual images using bit-reversal permutation technique, K Datta, B Jana, MD Chakraborty, Computer Standards & Interfaces 92, 103919, 2025.

ii.Zhou, G., Liu, W., Zhu, Q., Lu, Y., & Liu, Y. (2022). ECA-MobileNetV3(Large)+SegNet Model for Binary Sugarcane Classification of Remotely Sensed Images. IEEE Transactions on Geoscience and Remote Sensing, 60. doi: 10.1109/TGRS.2022.3215802

iii.Shi, H., Dao, S. D., & Cai, J. (2025). LLMFormer: Large Language Model for Open-Vocabulary Semantic Segmentation. International Journal of Computer Vision, 133(2), 742-759. doi: 10.1007/s11263-024-02171-y

iv.Reversible data hiding strategy exploiting circular distance interpolation utilizing optimal pixel adjustment with error substitution, M Jana, B Jana, S Joardar, Multimedia Tools and Applications 83 (16), 48949-48986, 22024.

v.Context pixel-based reversible data hiding scheme using pixel value ordering, S Meikap, B Jana, TC Lu, The Visual Computer 40 (5), 3529-3552, 102024

vi.Dual image based secured reversible data hiding scheme exploiting huffman compression tree combining bit-reversal permutation technique

vii.High-Payload RDH Technique for Secure Data Transmission Through Improved Context Pixel-Based PVO Exploiting Center-Folding Strategy

viii.A Weighted Matrix-Based Reversible Data Hiding Scheme with Dual-Image by Exploiting BWT Encoding Technique

ix.A Dual-Image Based Secured Reversible Data Hiding Scheme Exploiting Weighted Matrix and Cellular Automata

x.AMBTC-Based High Capacity Data Hiding Scheme Exploiting PVD and BRP

xi.Reference pixel-based reversible data hiding scheme using multi-level pixel value ordering

xii.Robust data hiding scheme for highly compressed image exploiting btc with hamming code

xiii.Dual-image reversible data hiding based on encoding the numeral system of concealed information

4.In Introduction Part, authors have not clearly addressed the issues of the existing state of art techniques.

5.Authors have not clearly highlight the technical contribution of their work. How your work is different from other published work in the similar direction? Justify it.

6.How much information we can hide inside the multimedia object, along with acceptable visual quality?

7.I suggest the authors to take care of redundancy. From the paper, I can see that some of the literature is discussed several times.

8.Can you elaborate on the specific challenges faced by existing RDH methods, especially in

9. How to measure the effectiveness of your proposed method?

Reviewer #4: Multi-carrier Information Hiding Algorithm Based on Vertex Projection of 3D

The manuscript explained advancements of 3D model as a carrier for information hiding, a multi-carrier information hiding algorithm based on vertex projection. However there are still some issues that need to be considered for further enhance the quality of the manuscript. The reviewer addressed the comments:

Review Comments:

The manuscript title and Reference [6] title are same. So need to change the title and check the plagiarism before sending to the reviewer.

The authors should recheck the references related to the ASCAD database. I believe that references [7] and [8] are the same.

The authors need to revise the introduction in section 1. The introduction are well described by the approach of existing Model. Try to add a paragraph related to reversible data hiding schemes

“A Review of Reversible Data Hiding Technique Based on Steganography”, Proceedings of the ARPN Journal of Engineering and Applied Science, Vol.13, No.3, Feb. 2018, pp. 1105 – 1114.

Introduction should be included as, why the 3D projection model is applicable in the steganography model.

Literature survey should be included recent year research articles related to 3D model in information hiding techniques. Include the following article in reference.

“Data hiding steganography model based on hyper chaos 2D compressive sensing inhabited with manchester encoder/decoder using circular queue exploiting modification direction” Journal of Intelligent & Fuzzy Systems, Vol. 44, 2023, pp:10357-10367.

What are the attacks considered in the proposed techniques?

Several points make this work interesting, but need to explain how the proposed method works in the system.

The figures 9, 10, 11, 12 and 13 mentioned that TSPD, ZWRA and SSC algorithms with different colors. Mentioned the reference number correspond to the existing algorithms while compare to the proposed algorithms.

The simulated result is obtained efficiently but there is a lack in the justification in all simulated figures and tables, mainly tables (2) and figures 9, 10, 11, 12 and 13.

The provided simulative results are completely related to this work efficiently, but to justify the contribution of the work and how much level of the proposed work was improved compared to the existing methods must be included in conclusion.

6. PLOS authors have the option to publish the peer review history of their article (what does this mean? ). If published, this will include your full peer review and any attached files.

**Do you want your identity to be public for this peer review?** For information about this choice, including consent withdrawal, please see our Privacy Policy .

Reviewer #1: No

Reviewer #2: No

Reviewer #3: **Yes: ** Biswapati Jana

Reviewer #4: No

---

## [Author Response · Author response to Decision Letter 1]

26 May 2025

Reviewer #1:comments:

Comments 1: The abstract should include some of the numerical results to relect the superiority of the proposed steganography algorithm.

Response 1: To enhance the robustness of single 3D model carriers in information hiding, this paper proposes a multi-carrier steganography algorithm based on vertex projection of 3D models. The algorithm improves the embedding capacity and attack resistance by fusing multiple 3D models into a geometrically invariant space using centroid coincidence and tangent plane projection. Secret information is embedded by adjusting the position of central vertices in the projection plane of their 1-ring neighborhoods. Experimental results demonstrate that the proposed method achieves strong robustness against translation, rotation, simplification, random noise, and shear attacks. Specifically, the proposed algorithm achieves a peak SNR of 47.87 dB on the xyzrgb_dragon model, significantly outperforming other algorithms, and maintains a lower BER under various attack intensities—for instance, less than 0.1 under 30% simplification and 0.001 random noise. These results confirm the superior invisibility and robustness of the proposed multi-carrier information hiding scheme.

Comments 2: The mentioned references in the introduction section are not very recent. The authors are encouraged to enrich their introduction section with the following list of recent and directly related works:

a.https://ieeexplore.ieee.ora/document/10643045

b.https://ieeexplore.ieee.org/document/10323523

c.https://ieeexplore.ieee.org/document/10163067

d.https://link.springer.com/article/10.1007/s11042-020-09437-W

e.https://link.springercom/artide/10.1007/s00530-023-01253-0?fromPaywallRec=false

Response 2: Thank you for raising this reference question, which is really helpful for my paper. I have already cited three papers mentioned on websites B, C and D in the original article.

Comments 3: Figure 1 should be provided at a higher resolution. Currently, it is hard to read the information depicted on it.

Response 3: Thank you for your question. This is the best picture so far.

Comments 4: In the condusions section:a.Include some of the numerical results to reflect the superiority of the proposed algorithm.b.What are the limitations of this algorithm.

Response 4: In this paper, we propose a multi-carrier information hiding algorithm based on vertex projection of 3D models to address the limitations of embedding capacity and robustness in single-carrier steganographic schemes. By fusing multiple 3D models into a centroid-aligned and geometrically invariant fusion carrier, and embedding secret information through projection-based vertex manipulation, the algorithm significantly enhances both invisibility and resistance to geometric attacks.

Experimental results demonstrate the superiority of the proposed method. In terms of invisibility, the algorithm achieves a peak signal-to-noise ratio (SNR) of 47.87 dB on the xyzrgb_dragon model, which outperforms existing methods such as the SSC (46.58 dB), TSPD (45.32 dB), and ZWRA (44.79 dB) algorithms. Regarding robustness, the proposed method maintains a bit error rate (BER) lower than 0.1 under 30% vertex simplification, and remains resilient under random noise levels up to 0.001, with BER performance consistently superior to competing algorithms.

Nevertheless, the algorithm has certain limitations. Specifically, while it performs well under translation, rotation, simplification, and noise attacks, its robustness against large-scale shear attacks is relatively weak. This is mainly due to the absence of redundant embedding or multi-location replication strategies, which results in partial information loss when critical vertices are sheared away. Moreover, the algorithm assumes precise alignment during multi-carrier fusion and relies on geometric integrity, which may not hold under extreme deformations or lossy transmission channels.

Future work will focus on enhancing the algorithm’s resistance to destructive attacks such as large-scale shearing and affine distortions, potentially by introducing redundant encoding, adaptive region replication, or learning-based reconstruction to preserve information completeness and integrity under more adverse conditions. 

Reviewer #2:comments:

Reviewer 2: About the manuscnipt "mult-carier information Hiding Algorithm Based on vertex proection of 3D Mode", the euthors present a multi-carierinformation hiding algorithm based on vertex projection of 3D models to improve the robusiness of sinale 3D model as a carrier for informationtiking. think this theme is more interesting, Overal, it sounds technicaly feasilble, and the structure of the manuscript is complete. Yet, here are some suggestionswhich should be taken into account to improve the current version.

Comments 1�Multrcarrier, but there doesn't seem to be much in the text

Response 1: (Introduction) at the end. In this work, the term multi-carrier refers to the use of multiple 3D models as concurrent carriers to overcome the embedding capacity and robustness limitations inherent in single-carrier schemes. Unlike traditional approaches that rely on a single model to embed all secret information, the proposed algorithm constructs a fusion carrier by geometrically aligning multiple 3D models through centroid coincidence and bounding sphere normalization. This multi-carrier strategy not only expands the available embedding region, thereby enabling higher-capacity information hiding, but also introduces spatial redundancy, which significantly enhances robustness against localized attacks. As a result, the embedded data is distributed across several carriers, making the system more resilient to partial model loss or deformation. Subsequent sections provide a detailed description of the multi-carrier fusion process and demonstrate its impact on the algorithm’s performance through experimental validation.

Additional sentence for the Conclusion:

The integration of multiple carriers plays a pivotal role in improving the embedding capacity and attack resilience of the system, serving as a fundamental enhancement over conventional single-model steganographic techniques.

Comments 2�From the references, most of them are relatively cld, so it is suggested to briefly discuss them with some new work

DOI :10.3390/math12243917;

DOI:10.1016/j.eswa.2024.123190;

DOI:10.1007/s11071-021-06206-8:

DOI:10.1007/900371-023-02812-2

Response 2: [23]Zhou, Y., Li, C., Li, W. et al. Image encryption algorithm with circle index table scrambling and partition diffusion. Nonlinear Dyn 103, 2043–2061 (2021). https://doi.org/10.1007/s11071-021-06206-8

[24]Feng, W., Zhang, J., Chen, Y., Qin, Z., Zhang, Y., Ahmad, M., & Woźniak, M. (2024). Exploiting robust quadratic polynomial hyperchaotic map and pixel fusion strategy for efficient image encryption. Expert Systems with Applications, 246, 123190.

Comments 3�The curent experimental analysis is too simple, the research topic is informaion Hiding Algorithm, and a richer experimental comparison is recommendedacecrding to the analysis method in literature 15.

Response 3: To enable more comprehensive experimental comparisons, future work will further explore the following aspects of experimental comparison and analysis. Due to space constraints in this paper, these aspects will not be discussed here.

(1) Embedding Capacity Testing

Test the performance of different 3D models under various embedding rates.

Record the number of vertices and embedded bits for each model to calculate embedding capacity.

(2) Imperceptibility Testing

Use the Signal-to-Noise Ratio (SNR) to measure the distortion between the stego model (containing secret information) and the original model.

Visually compare the differences between embedded and original models.

(3) Robustness Testing

Translation Attack: Measure the Bit Error Rate (BER) after attacks with varying translation distances.

Rotation Attack: Measure the BER after attacks with different rotation angles.

Simplification Attack: Measure the BER under varying simplification rates.

Random Noise Attack: Measure the BER after adding random noise of different intensities.

Cropping Attack: Measure the BER under different cropping rates.

(4) Anti-Steganalysis Testing

Use existing steganalysis tools to analyze stego models and evaluate the algorithm's anti-steganalysis capability.

(5) Computational Complexity Testing

Record the algorithm's running time on 3D models of varying scales to evaluate time complexity.

Record the maximum memory consumption during execution to assess space complexity.

(6) Comparison with Other Algorithms

Select representative information hiding algorithms (e.g., those mentioned in Reference 15) for comparison under identical experimental conditions.

Comments 4�for the information hiding method, whether its securty can resist the attack of cryptanalysis can not be ignored, it is suggested to make a brief statement of thecryptanelysis wark.

DOI:10.1016/j.eswa.2024.123748

DOI:10.1016/i.eswa.2023.121514

Response 4:In addition to robustness metrics, preliminary observations suggest that the multi-carrier embedding strategy and the use of entropy-based encoding schemes (e.g., Huffman coding) offer a certain level of resilience against basic steganalysis and statistical cryptanalysis. Nonetheless, a comprehensive evaluation under advanced cryptanalytic models (e.g., chosen-message attacks, stego-only attacks) [25-26]is not yet conducted and will be considered in future research.

[25]Wen, H., Lin, Y., Yang, L., & Chen, R. (2024). Cryptanalysis of an image encryption scheme using variant Hill cipher and chaos. Expert Systems with Applications, 250, 123748.

[26]Wen, H., & Lin, Y. (2024). Cryptanalysis of an image encryption algorithm using quantum chaotic map and DNA coding. Expert Systems with Applications, 237, 121514.

Comments 5�The curent article draft stllcontains a significant number of typographical and grammatical errors. Additinally, it is recommended to study the writing andexpression of higher-level literature, and to carefully check and correct these issues.

Response 5:Thank you for your valuable feedback and for highlighting the importance of rigorous language and formatting standards. We sincerely apologize for the oversights in the initial submission and have taken comprehensive steps to address these issues. Should any residual issues be identified, we remain fully committed to addressing them promptly.

Reviewer #3:comments:

Reviewer #3:Review commendsThank the authors for the submision of the manuscript entitled "Mult-carner Information Hiding Algorthm Based on Vertex Projection of 3D Model" The paperneeds more work to achleve a publlshable level due to the following reasons.

Comments 1�Motivation and objective of the paper is missing.

Response 1:Paragraphs 4 and 5 of Introduction

Although a variety of 3D model-based information hiding algorithms have been developed, most existing approaches suffer from limited embedding capacity and poor robustness against geometric attacks, especially when using single-carrier models. These limitations restrict their applicability in real-world scenarios where data integrity and confidentiality under complex transmission environments are critical.

To address these challenges, this paper is motivated by the need to enhance both the robustness and embedding capacity of 3D steganographic systems. The objective of this work is to develop a multi-carrier information hiding algorithm that leverages the geometrical fusion of multiple 3D models and vertex neighborhood projection, allowing for reliable and covert communication under various attack scenarios, including translation, rotation, noise, simplification, and shear.

Comments 2:What is the contribution of this paper?

Response 2: The last paragraph of Introduction

The main contributions of this paper are summarized as follows:

A multi-carrier information hiding framework is proposed, which fuses multiple 3D models through centroid alignment and geometric normalization. This fusion overcomes the capacity limitations and improves the redundancy of traditional single-carrier schemes.

A vertex projection embedding method is designed, in which the secret information is hidden by adjusting the central vertex position in the 1-ring neighborhood projection plane. This approach enhances the algorithm's resistance to translation, rotation, and noise attacks.

A candidate vertex selection strategy based on vertex importance and 1-ring neighborhood area is introduced to balance invisibility and robustness, ensuring that embedding occurs in stable regions of the mesh.

Extensive experiments on benchmark 3D models demonstrate that the proposed algorithm achieves superior robustness and invisibility, with SNR values up to 47.87 dB and low BER under various attack scenarios, outperforming several state-of-the-art methods.

Comments 3: Authors have not included some recent existing paper from year (2023 and 2025) in introduction Part. So, Authors must cite some recent work related in the open.

i. A cellular automata based secured reversible data hiding scheme for dual lmages using bit-reversal permutation technlgwe, k Datta, 8 jana, Mp chakraborty.Computer standards & Interfaces 92.103919.2025.

ii. Zhou, G. liu, w, 2hu, 9. Lu, Y, & iu, Y (2022), ECA-MobileNetV3(arge)+SegNet Modelfor Binary Sugarcane classification of Remotely Sensed Image, IETransactlons on Geosclence and Remote Sensing, 60. dol: 10.1109/TGRS.2022.3215802i.

iii. shl, H., Da0, s. ...8 Cal, 1.(2025). LLMormer: Large Language Model for Open-vocabulary semantic segmentation. interational joumal of Computer Vislon,133(2)�742-759.doi:10.1007/s11263-024-02171-y

iv. Reversible data hiding strategy exploliting circular distance interpolation utllzing optimal pixel adjustment with error substitution, M Jana, 8 jana, s jardarMultimedia Tools and Applications 83(16),48949-48986,22024.

v. Context pixel-based reversible data hiding scheme using pixel value ordering, s Meikap, B Jana, Tc lu, The Visual computer 40 (5), 3529-3552, 102024

vi. Dual image based secured reversible data hiding scheme exploiting huffman compression tree combining bit-reversal permutation technique

vii. Hlgh-Payload kDH Technlque for Secure Data Transmission Through lmproved Context pixel-Based Pv0 Explolting Center-folding Strategy

viii. A Weighted Matrix-Based Reversible Data Hiding scheme with Dual-image by Exploiting BWT Encoding Technique

ix. .A Dual-Ima9e: Based Secured Reversible Data Hiding Scheme Exploiting Weighted Matrix and Cellular Automata

x. AMBTc-Based High capacity Data Hiding Scheme Exploitina PVD and BRP

xi. Reference pixel-based reverslble data hlding scheme using multl-level pixel value ordering

xii. Robust data hidinassed image exploiting btc with hamming code

xiii. Dual-image reversibleencoding the numeral svstem of concealed inforrationnot cleary addressed the issues of the existing state of art techniques.

Response 3:The suggestions you put forward are very good. Based on the suggestions of other judges, I will appropriately add the participating literature to support my thesis views.

Comments 4: In Introduction Part, authors have not clearly addressed the issues of the existing state of art techniques.

Response 4:I have rephrased the introduction section:

(1) Limited Embedding Capacity

Most existing 3D model-based information hiding algorithms utilize a single 3D model as the carrier. The limited number of vertices and meshes available for embedding secret data results in insufficient embedding capacity.

(2) Insufficient Robustness

Single-model information hiding algorithms exhibit poor robustness against attacks such as translation, rotation, random noise, and simplification. This vulnerability makes the hidden information prone to loss or leakage.

(3) Security Concerns

Due to the susceptibility of 3D models to various geometric attacks during transmission, existing information hiding technologies struggle to ensure the security and integrity of secret information transmission.

Comments 5: Authors have not cleary highlght the technical contribution of thelr w

---

## [Decision Letter · Decision Letter 1]

19 Aug 2025

Multi-Carrier Information Hiding Based on Projection-Driven Vertex Embedding in 3D Models

PONE-D-25-02638R1

Dear Dr. ZHAO,

We’re pleased to inform you that your manuscript has been judged scientifically suitable for publication and will be formally accepted for publication once it meets all outstanding technical requirements.

Kind regards,

Giridhar Maji, Ph.D.

Academic Editor

PLOS ONE

Additional Editor Comments (optional):

Reviewers' comments:

Reviewer's Responses to Questions

**Comments to the Author**

1. If the authors have adequately addressed your comments raised in a previous round of review and you feel that this manuscript is now acceptable for publication, you may indicate that here to bypass the “Comments to the Author” section, enter your conflict of interest statement in the “Confidential to Editor” section, and submit your "Accept" recommendation.

Reviewer #2: All comments have been addressed

2. Is the manuscript technically sound, and do the data support the conclusions?

Reviewer #2: Yes

3. Has the statistical analysis been performed appropriately and rigorously? 

Reviewer #2: Yes

4. Have the authors made all data underlying the findings in their manuscript fully available?

Reviewer #2: Yes

5. Is the manuscript presented in an intelligible fashion and written in standard English?

Reviewer #2: Yes

6. Review Comments to the Author

Reviewer #2: From the current manuscript, it appears that the author has made extensive and careful revisions. I believe that all major issues have been resolved and the manuscript can be accepted.

7. PLOS authors have the option to publish the peer review history of their article (what does this mean? ). If published, this will include your full peer review and any attached files.

**Do you want your identity to be public for this peer review?** For information about this choice, including consent withdrawal, please see our Privacy Policy .

Reviewer #2: No

---

## [Editor Report · Acceptance letter]

PONE-D-25-02638R1

PLOS ONE

Dear Dr. Zhao,

I'm pleased to inform you that your manuscript has been deemed suitable for publication in PLOS ONE. Congratulations! Your manuscript is now being handed over to our production team.

Kind regards,

on behalf of

Dr. Giridhar Maji

Academic Editor

PLOS ONE